# Cyclic Stretch-Induced Mechanical Stress Applied at 1 Hz Frequency Can Alter the Metastatic Potential Properties of SAOS-2 Osteosarcoma Cells

**DOI:** 10.3390/ijms24097686

**Published:** 2023-04-22

**Authors:** Giulia Alloisio, David Becerril Rodriguez, Marco Luce, Chiara Ciaccio, Stefano Marini, Antonio Cricenti, Magda Gioia

**Affiliations:** 1Department of Clinical Sciences and Translational Medicine, University of Rome ‘Tor Vergata’, Via Montpellier 1, I-00133 Rome, Italy; giulia.alloisio@alumni.uniroma2.eu (G.A.); chiara.ciaccio@uniroma2.it (C.C.); stefano.marini@uniroma2.it (S.M.); 2Institute of Structure Matter del Consiglio Nazionale delle Ricerche ISM-CNR, Via del Fosso del Cavaliere 100, I-00133 Rome, Italy; d.becerril.r@gmail.com (D.B.R.); marco.luce@artov.ism.cnr.it (M.L.); antonio.cricenti@artov.ism.cnr.it (A.C.)

**Keywords:** osteosarcoma, malignant bone cells, cell morphology, cell structural deformation, cell migration, mechanical stress, biomechanical response, mechanobiology, cyclic mechanical strain, mechano-sensing

## Abstract

Recently, there has been an increasing focus on cellular morphology and mechanical behavior in order to gain a better understanding of the modulation of cell malignancy. This study used uniaxial-stretching technology to select a mechanical regimen able to elevate SAOS-2 cell migration, which is crucial in osteosarcoma cell pathology. Using confocal and atomic force microscopy, we demonstrated that a 24 h 0.5% cyclic elongation applied at 1 Hz induces morphological changes in cells. Following mechanical stimulation, the cell area enlarged, developing a more elongated shape, which disrupted the initial nuclear-to-cytoplasm ratio. The peripheral cell surface also increased its roughness. Cell-based biochemical assays and real-time PCR quantification showed that these morphologically induced changes are unrelated to the osteoblastic differentiative grade. Interestingly, two essential cell-motility properties in the modulation of the metastatic process changed following the 24 h 1 Hz mechanical stimulation. These were cell adhesion and cell migration, which, in fact, were dampened and enhanced, respectively. Notably, our results showed that the stretch-induced up-regulation of cell motility occurs through a mechanism that does not depend on matrix metalloproteinase (MMP) activity, while the inhibition of ion–stretch channels could counteract it. Overall, our results suggest that further research on mechanobiology could represent an alternative approach for the identification of novel molecular targets of osteosarcoma cell malignancy.

## 1. Introduction

Most cell responses that have been biochemically characterized are specifically activated following ligand–receptor binding. However, mechanobiology studies have shown that cells can alternatively trigger specific biochemical pathways via a multitude of mechanical stimuli [1]. Indeed, external biomechanical forces and matrix mechanics have been reported to be critical regulators of stem cell fate [2,3]. The application of loading forces (e.g., cyclic tensile forces) has been reported to promote anabolism by stimulating osteogenesis both in vitro and in vivo [4,5,6]. Interestingly, mechanical factors are essential not only for the well-known preservation of bone quality and quantity but also because they contribute to the development of cancer [7,8,9,10]. Increasing evidence suggests that environmental mechanics has a significant impact on gene expression, cell fate, and function and ultimately gives rise to modifications of the malignancy of osteosarcoma [7,11,12].

Osteosarcoma (OS) is the most common primary malignant bone tumor in children and adolescents, associated with locally aggressive growth and a high risk of metastases in early-stage cancer. During their metastatic journey, cancer cells are exposed to several biophysical challenges. In fact, they acquire genetic and epigenetic modifications that render them more competitive in the neoplastic microenvironment [13,14]. For instance, malignant cells show increased migration, reduced adhesion, and higher compliance [3].

Most of the targeted therapies for osteosarcoma have focused on signaling pathways such as human epidermal growth factor receptor (HEGR2), insulin-like growth factor 1(IGF1), and mammalian target of rapamycin (mTOR), which appear overactive in osteosarcoma tumors [12,15]. Current data demonstrate that commonalities in cell responses to different types of mechanical stimuli are correlated with the differential activation of intracellular signaling [16]. Regarding mechano-oncology, a few biophysical markers of malignancy have been discovered to be related to the mechanical forces they were subjected to [17,18]. Nonetheless, targeting molecules involved in mechano-transduction is believed to have promising innovative potential because the identification of mechano-sensor molecules, which regulate osteosarcoma cell pathology, could reveal new oncogenic targets that are not identified in conventional biochemical screening for chemotherapy [11,19,20].

Following a number of studies regarding the effect of environmental mechanical cues on cell volume regulation, changes in cell volume are considered part of the mechanical signal mechano-transduction process for chondrocyte and mesenchymal cells [20,21]. Furthermore, dynamically applied mechanical forces have been reported to produce modifications in cellular shape [22], and a stretch-induced elongated shape in mesenchymal cells can aid osteogenic differentiation [4,23]. Previous results concerning biomechanical changes have reported that low metastatic osteosarcoma cells display larger spreading sizes and generate higher forces than the highly malignant variants [17]. Notably, a variation in cell volume can modify a cell’s ability to move and thus hinder cell migration during development and tumor progression [24].

In this study, atomic force microscopy (AFM) and confocal microscopy, which are leading imaging techniques used in 3D cellular morphology, were used for an in vitro study of the mechanobiological responses of osteosarcoma SAOS-2 cells. Since during fast-walking activities bone cells are subjected to moderate elongation [25], we hypothesized that cells might evolve a mechano-biological response to a physiological stretch. Thus, our culturing experimental model was stimulated with a medium-magnitude stretch (corresponding to 0.5% elongation). Ab initio, we screened for the frequency (Hz) that worked best in applying the 0.5% elongation cyclic in order to induce the widest biological response of SAOS-2 cells (for further information, see the Appendix A). Moreover, with cell shape being affected by the balance between external and internal forces, it was theorized that applying cyclic tensile forces to adherent cells would prompt a consequent change in cell morphology.

Since the enlargement of the nucleus (due to the increased amounts of chromatin present within malignant cells) and increases in the nuclear-to-cytoplasmic (N/C) ratio have been reported to be correlated with an increase in cell malignancy [26,27,28], we looked for stretch-induced phenotypical changes by measuring the classical AFM morphological parameters (i.e., height, area, shape, and roughness) of the stretched and untreated SAOS-2 cells. Next, given that nuclear size changes may potentially impact gene expression through the modulation of intranuclear structures, we hypothesized that stretch-induced morphological changes could result in a variation of the osteoblastic differentiative grade. Therefore, the levels of the osteogenic gene markers (namely, *RUNX*-*2*, *COL1A1*, and *ALPL*) of 24 h 1 Hz-treated and untreated SAOS-2 cells were measured via quantitative real-time PCR (polymerase chain reaction).

Finally, we assessed the induced changes in cellular behavior by looking at the osteoblastic adhesiveness, the migration, and the transmigration (using a wound healing assay and a trans-well Boyden chamber) of pretreated or untreated SAOS-2 cells with the 24 h 1 Hz stimulation.

Since mechanical stresses from the microenvironment have been reported to regulate the PI3K/Akt signaling pathway (the most activated downstream effector in the oncogenic landscape) and the MMP-mediated degradation of the extracellular matrix [29,30,31,32], we decided to investigate whether these were sensible or not for our mechanical stimulus. Furthermore, cell volume regulation, which plays a role in cell migration, has been reported to be rapidly achieved through the participation of ion transportation across the cell membrane [33,34].

To select the class of molecules involved in the mechanism underlying the stretch-induced upregulation process, we tested whether or not soluble inhibitors could counteract the stretch-induced migration of SAOS-2 cells. Hence, considering that stretch-induced changes in cell motility could depend on force-sensitive ion-channels and/or MMP’s proteolytic activity, we screened MMP’s broad spectrum inhibitor and ion stretch-channel blocker.

## 2. Results

### 2.1. The 24 h 1 Hz Cyclic Stretch-Induced Increase in the Nuclear Surface of SAOS-2 Cells

In order to understand the cell biology of the osteosarcoma response to mechanical forces, SAOS-2 cells were cultured on a deformable silicone plate system and then were exposed or not to a 0.5% elongation uniaxial cyclic stretch at 1 Hz frequency for 24 h (see Appendix A for information regarding the stimulus selection).

In agreement with the literature, the measurements of nuclear heights using the AFM of SAOS-2 cells on a glass coverslip were found to be 1.171 ± 0.230 µm (Appendix A). A 24 h 1 Hz stretch pretreatment did not significantly change the nuclear height (1.062 ± 0.233 µm, Appendix A). To look for stretch-induced phenotypical changes, we started the morphological analysis with the comparison of the nuclei of SAOS-2 cells between stretched and untreated specimens (plated on the silicone plate). The identification of cell surface areas, referred to as nucleus and non-nucleus, was performed as described in the Materials and Methods section. The upper panel of Figure 1 illustrates representative pictures from the AFM microscopy of the nuclear profile of SAOS-2-adherent cells fixed on the silicone plate either stretched for 24 h or not (Figure 1B and Figure 1A, respectively). The average root mean square of the profile irregularities of the nuclei cultured on the silicone plates was found to be 0.655 ± 0.163 µm for the height and 0.215 ± 0.081 µm for roughness (Figure 1C,D). Given that a 24 h 1 Hz cyclic stretch did not change the height or roughness of nuclei, we looked for stretch-induced changes by measuring the area and the shape (eccentricity) of SAOS-2 nuclei.

The minor and major axes of the nuclei were measured via AFM to derive the eccentricity (E) and the area of the nuclei according to Equations (1) and (2), respectively.
(1)$$\mathrm{eccentricity},\;E=\frac{\mathrm{minor}\;\mathrm{axis}}{\mathrm{major}\;\mathrm{axis}}$$
(2)$$\mathrm{nucleus}\;\mathrm{area},\;A(nucleus)=\frac{\prod_{}^{}minor\;axis\cdot major\;axis}{4}$$

Figure 1E clearly shows that the 1 Hz mechanical treatment increased the nuclear area of treated samples compared with the unstretched cells. Specifically, the mean nucleus area of control samples, 199.5 ± 63.37 µm^2^, was found to be 1.30-times smaller than the mean nucleus area of 1 Hz stimulated cells (i.e., 258.8 ± 48.72 µm^2^) (*p* < 0.05). To assess whether or not the nuclear size variation was accompanied by a deformation of the nuclear shape, Equation (1) was applied, and the nuclear eccentricity was found to be 0.6, which was similar in stretched and unstretched cells (Figure 1F). Student’s *t*-test showed that the eccentricity average values of the two groups did not differ (*p* > 0.05), indicating that the mechanical stimulation induced an increase in the size of the nucleus without interfering with its oval shape.

### 2.2. The 24 h 1 Hz Cyclic Stretch-Induced Phenotypical Changes in SAOS-2 Cells

The upper panel of Figure 2 illustrates representative confocal microscopy pictures. The lower panel of Figure 2 reports the bar plots of measurements detected for the mechanically stimulated and unstimulated cells. Similar to the nucleus (Figure 1E), the entire cell also increased its surface area (Figure 2A). The measurements of whole cell areas showed that the 1 Hz cyclic deformation applied for 24 h induced a doubling of the cell surface area to a value of 2962 µm^2^ when compared with the unstretched counterparts (1499 µm^2^) (*p* < 0.01; difference between means ± SEM: 1463 ± 412.0) (Figure 2A).

Different from the variation in nuclear size, the variation in cell area was accompanied by a deformation of the whole cell body, which became more elongated following the cyclic stretch application. Figure 1 and Figure 2 clearly show that, different from the nuclear profile, the whole cell profile cannot be adequately described simply as an oval shape (as it does not display an elliptic geometry). Therefore, for the measurement of the whole cell shape elongation, in addition to the eccentricity (E), the roundness (R) index was also calculated (Equation (3)) as an additional element in the measurement of rotundity, as previously described [22].
(3)$$\mathrm{roundness},\;R=\frac{4\;A}{\prod \cdot {\left(major\;axis\right)}^{2}}$$

The rounder the shape of the cell is, the more the coefficients E and R approach a value of one (Equations (1) and (3), respectively). Overall, whether or not the silicone culture system was 1 Hz mechanically stimulated, the SAOS-2 cells presented an elongated cell body, expressed as eccentricity (E) and roundness (R) indexes (specifically, they were lower than 0.5; Figure 2B,C). Interestingly, following the 24 h 1 Hz cyclic mechanical treatment, the eccentricity and roundness were further reduced by 40%, *p* < 0.001 and *p* < 0.01, respectively (Figure 2B,C).

This mechanically induced cell elongation did not align the cells in a preferential direction. Appendix A clearly demonstrates that the elongated whole cell body displayed the major axis of the cell body as randomly oriented compared with the *x*-stretch axis (see Appendix A).

In an attempt to quantify stretch-induced variations in the membrane processing of the SAOS-2 cells (clearly visible in Figure 1 and Figure 2), the circularity (C) parameter was derived according to Equation (4) and considered an indicator of the star-like degree of the cell profile (Figure 2D),
(4)$$\mathrm{circularity},\;C=\frac{4\cdot \prod \cdot A}{{\left(perimeter\right)}^{2}}$$

The more the cell profile assumes a star-like shape, the more coefficient C approaches a value of zero (Equation (4)). The small protuberances radically increase the perimeter, which, in turn, increases the circularity measurement geometrically. Specifically, it appears that circularity may be a good indicator of the presence of pseudopodia because it decreases significantly as the number of small protrusions increases. Figure 2D shows that the 1 Hz mechanical stimulation did not significantly change the star-like profile of cells when comparing unstretched controls with cyclically stretched ones. Similarly, by counting the amount of cell processing per cell in confocal microscopy pictures, no significant differences could be detected between treated and untreated samples (Figure 2E). Although the mean of pseudopodia per cell was slightly higher than for untreated cells (0.41 pseudopodia per cell were counted in the mechanically stimulated sample, whereas 0.32 pseudopodia per cell were counted for the static control cells), this difference was not statistically significant (Figure 2E).

A confocal microscopy technique was also employed to investigate whether or not mechanical stimulation could disturb the N/C cell scaling. Therefore, the estimation of the nuclear–cytoplasmic ratio, N/C, was compared by measuring (i) the elliptical areas of the nuclei directly, whereas (ii) the size of the cytoplasm was indirectly derived by subtracting the nuclear area from the measurement of the whole cell surface. Figure 2F shows that the 24 h cyclic stretch in the SAOS-2 cells induced a disruption of the cell scaling N/C, which decreased by 22% (*p* < 0.05).

### 2.3. The 24 h 1 Hz Cyclic Stretch-Induced Changes in the Cell-Surface Peripheral Roughness

Although the AFM analysis regarding the height, roughness, and area of in- and off-nucleus regions was found to not be statistically different (Figure 1C–E and Appendix A), a qualitative difference in the complexity of the cell boundary ultra-structures could be seen in the treated cells compared with the untreated cells (Appendix A). We hypothesized that the forces on the cell membrane would also affect the cell periphery and the local cell protrusions (see Appendix A). To verify our hypothesis, we first defined the cell periphery area as the region within 2 μm of the cell membrane boundary, as shown in Figure 3, and then, we performed the AFM analysis, restricting it to the thick perimeter (Figure 3A,B and Appendix A). Interestingly, significant quantitative changes were obtained by restricting the surface ultrastructure analysis to the cell periphery. The comparative analysis showed that the average roughness of the cell periphery increased by 40% in the mechanically treated sample, with the following mean values: 0.161 ± 0.043 μm and 0.232 ± 0.091 μm for the control and 1 Hz treated sample, respectively (Figure 3C).

### 2.4. The 24 h 1 Hz Cyclic Stretch-Induced Morphological Variation Did Not Correlate with Changes in the Cell Viability and the Differentiation Grade of SAOS-2 Cells

An in-house-developed technical approach allowed us to make absorbance and fluorescence readings directly for live cells within the silicone well plate shortly after the mechanical stimulation (for further details, see Appendix A). Figure 4A shows that the 24 h 1 Hz stimulus did not affect SAOS-2 cell viability, as displayed by the absorbance measurements, which reflect the mitochondrial metabolic activity (using the MTS soluble probe). Given that indirect measurements of the number of viable cells using the mitochondrial metabolic rate have been reported to possibly be inaccurate in several different experimental conditions [35], we also validated that, effectively, there were no differences in viable cell numbers by looking at the DNA cell content through Cy Quant fluorescent dye. Although Figure 4B shows a slight increase in the DNA cell content of the treated sample compared with the control counterpart, this change was not significantly different. The data recorded on the silicone well plate were in agreement with the results recorded within conventional 96-well plate systems (Appendix A).

Furthermore, as Alkaline phosphatase (ALP) activity functionally mirrors the differentiation grade of osteoblastic cells and is significantly associated with the presence of metastasis in osteosarcoma patients [36], we measured the osteoblastic-specific functionality. The measurements of the ALP activity on the silicone well plate using the Blue ALP kit demonstrated that the ALP enzymatic activity is not modulated by the 24 h 1 Hz uniaxial stimulation, as no meaningful differences were detected between stimulated and control static cells (Figure 4C). The absence of ALP activity variation was also validated by measuring the ALP activity on a 96-well plate using fluorogenic peptide (Appendix A).

To verify whether or not the mechanically induced morphological changes measured at 24 h (Figure 1E, Figure 2A–C,F and Figure 3C) were correlated to the deregulation of bone marker genes, the gene expression of early and late differentiation markers *ALPL*, *RUNX-2*, *COL1A1*, and *OCC* was compared with the 24 h 1 Hz stretched and untreated SAOS-2 cells. The Materials and Methods section reports the primer sequences used for the amplification of the bone differentiative biomarker genes: runx-2 (*RUNX-2*), collagen α1 (I) chain (*COL1A1*), alkaline phosphatase (*ALPL*), and osteocalcin (*BGALP/OCC*). The housekeeping *GAPDH* enzyme was employed for gene normalization. It came as no surprise that the expression levels of the osteocalcin (*BGALP/OCC*) gene could not be quantified because the basal expression levels were found to be too low for a late-differentiation marker. The analysis of gene expression did not show any significant changes in the gene expression of *ALPL*, *RUNX-2*, or *COL1A1* between SAOS-2 cells treated or untreated with a 24 h 1 Hz cyclic stretch (Figure 4D).

### 2.5. Mechanical Cell Pre-Conditioning Promotes Cell Motility and Reduces Adhesion of SAOS-2 Cells

To check whether the 24 h 1 Hz cyclically induced morphological changes correlated with modifications in cell behavior, we looked for stretch-induced changes in cell motile properties. SAOS-2 cells were exposed, first to 1 Hz cyclic stretching with a 0.5% elongation for 24 h, and then, migration and transmigration were assessed to characterize the stretch-induced cell motility changes using wound healing and trans-well Boyden chamber assays, respectively. Figure 5A shows that cyclic uniaxial stretching significantly promoted migration because, after 20 h of migration, pre-stretched SAOS-2 cells reached 20% of closure, while untreated control cells did not migrate and increased the wound gap by 4% (*p* < 0.001).

The effect of mechanical stimulation on cell transmigration is consistent with these findings. The cell transmigration ability was recorded by observing the number of cells able to pass through the Boyden chamber. As presented in Figure 5B, 24 h after seeding, mechanically pre-stimulated SAOS-2 cells showed a five-fold increase in transmigration capacity compared with static control cells (*p* < 0.0001).

To determine whether the cell adhesion capability was affected by the 1 Hz mechanical stress after the 24 h stimulation, SAOS-2 cell adhesive capacity was evaluated on type-I-collagen-coated conventional dishes, and the number of attached cells was compared with treated and untreated cells through two concurrent measurements: cell confluence and the number of cells (Figure 5C and Figure 5D, respectively). The SAOS-2 cell adhesion capability decreased in the mechanically stimulated samples compared with their untreated counterparts, with a significant difference in the two methods (*p* < 0.001 and *p* < 0.0001; see Figure 5C and Figure 5D, respectively). Importantly, the cells of Figure 2 and Figure 5 do not share the same cell timing, nor do they adhere to the same type of support (for further details see Appendix A). Thus, the increase in cell area induced by the mechanical stimulation (Figure 2A) does not contradict the data of Figure 5C,D, which show that the 24 h 1 Hz treatment decreased the SAOS-2 cell’s ability to adhere to conventional culturing conditions in vitro.

### 2.6. The 24 h 1 Hz Cyclic Stretch-Induced Cell Motility Upregulates the Activity of Soluble MMP-2 with No Perturbation of the Protein Levels of Akt

We performed a zymography assay to evaluate the presence of secreted MMPs in the conditioned media of the SAOS-2 cells. Figure 6A displays the zymogram of the gelatinolytic activity of the SAOS-2 secretome from the 24 h stretched cells and their control counterpart. The densitometric analysis indicated an upregulation of the catalytic activity of pro-MMP-2 induced by the stretch stimulation compared with the static control counterpart (*p* < 0.001; difference between means ± SEM, 18,789 ± 1515; see Appendix A).

To verify whether the observed MMP-2 upregulation was the downstream effect of the mechanical activation of the PI3K/Akt signaling pathway, we compared the total levels of Akt protein in stretched or untreated SAOS-2 cells via Western blotting (Figure 6B). The quantitative densitometric analysis of the detected bands displayed no significant difference between mechanically treated and control lysates (Appendix A). Similarly, the levels of the phosphorylated Akt form (60 kDa) were also not changed by the 24 h 1 Hz stimulation (Figure 6C).

### 2.7. The Cationic Channel Blocker, GsMTx4, Was Able to Counteract the Mechanical Stretch-Induced Cell Migration

To understand the mechanism that could underly the stretch-induced upregulation of SAOS-2 migration, we tested whether or not the stretch-induced migration could be counteracted by MMP’s inhibitors or by ion-channel blockers. The migration or transmigration ability was measured using wound healing and Boyden chamber assays in the presence or absence of the specific soluble inhibitors (Figure 7).

Both transmigration and wound healing assays were performed with or without 1.6 µM ilomastat (a covalent broad-spectrum inhibitor of the MMP active site). The quantitative analysis of both approaches showed that ilomastat fails to revert the stretch-induced upregulation of cell migration (Figure 7A–D).

To validate whether the cell motility induced by the 24 h 1 Hz stretch could be mediated by the stretch activation of cation channels, we employed GsMTx4 (a specific mechanosensitive and stretch-activated ion-channel inhibitor).

Therefore, SAOS-2 cells were stimulated with a 24 h 1 Hz cyclic stretch in the presence or absence of either 1 µM GsMTx4, and the migration or transmigration ability was compared using wound healing and Boyden chamber assays, respectively (Figure 7A–D).

Figure 7A shows that 1 µM GsMTx4, a specific mechanosensitive ion-channel inhibitor, was effective in counteracting cell-migrative upregulation (decreasing by 33.5% in the mechanically treated sample; *p* < 0.001, n = 12). In agreement with these findings, Figure 7C shows that GsMTx4 was the sole compound able to slow down the stretch-induced up-regulation of cell transmigration (decreased by 50% in the mechanically treated sample; *p* < 0.05, n = 4).

## 3. Discussion

The mechanical properties of cells that depend on cytoskeleton architecture have been reported to play a critical role in the mechano-transduction process and have great potential in cancer diagnosis and therapy [37]. Cell shape emerges from a balance between internal and external forces propagated through the cytoskeleton, the cell membrane, and cell–substrate adhesions. Even though many of these constituent elements have been studied in great detail at the molecular level, the mechanism by which global morphology is molded to address cell motility has yet to be fully understood.

This study analyzes stretch-induced changes in the morphological and motile properties of osteosarcoma cell biology. SAOS-2 cells were assessed using atomic force microscopy, confocal microscopy, and cell-based assays. As cyclic mechanical strain has been reported to affect osteosarcoma cell biology depending on the frequency by which the stretch is applied to cells [38], we started the investigation by screening the best frequency capable of inducing the widest biological response. Our results demonstrated that a 1 Hz frequency of mild mechanical stretching (corresponding to 0.5% Ɛ elongation) applied for 3 days was able to increase the proliferation rate (Appendix A). To characterize how the SAOS-2 cells respond to 1 Hz mechanical stimulation, the cell response was investigated in depth in a shortened process of one day. Our data demonstrated that changes in the proliferation rate following a 3-day stimulus were not significantly induced after a shortened treatment of 1 day (Appendix A). However, the 24 h application of the 1 Hz cyclic tensile forces induced morphological changes in both the shape and dimension of the nucleus and cytoplasm (Figure 1 and Figure 2), which was accompanied by the elongation of the whole cell body (Figure 2B,C), without inducing any preferential orientation in the cells (Appendix A). As most cell types orient themselves perpendicularly to (or along) the direction of uniaxial stretches when treated with high strain levels [39,40], we cannot exclude that SAOS-2 cells can orient themselves in response to prolonged mechanical stimulations and the elongation magnitude (Ɛ).

Metastatic cancer cells, in many cancers, have been reported to have altered cytoskeletal properties. In particular, the cells become more deformable and contractile. Consequently, the shape characteristics of more metastatic cancer cells would be expected to diverge from those of their parental cells [41]. Interestingly, large and round tumor nuclei in osteosarcoma have been reported to be correlated with good clinical outcomes [26]. In line with measurements reported in the literature for osteosarcoma cells [11,27], under our experimental conditions, the nuclear shape of SAOS-2 also displayed an elliptical geometry with an E-value of around 0.6. A comparison of cell morphology between cells that were or were not subjected to the 24 h 1 Hz treatment showed that the typical oval shape of osteosarcoma nuclei was retained, with the starting nuclear surface becoming even larger following mechanical stimulus (Figure 1).

In addition, our quantitative morphological analysis demonstrated that the 24 h 1 Hz mechanical stimulation could decrease the nuclear-to-cytoplasmic (N/C) ratio of the cyclically stretched SAOS-2 cells by 22% when compared with unstretched cells (Figure 2F). Interestingly, osteosarcoma cytoplasm changes in size have been reported to be correlated with increased metastasis, and the nuclear-to-cytoplasmic (N/C) ratio is believed to be essential for proper cell function [26,27]. Since the morphological abnormalities of the nucleus and N/C cell scaling are essential diagnostic features needed to distinguish benign and malignant cells in the daily practice of cytopathology [28], we hypothesize that 24 h 1 Hz exogenous stimulation could be exploited to identify the cellular pathways underlying biological abnormalities in malignant cells. However, to gain insights into genetic and epigenetic abnormalities, further studies are needed regarding the 24 h 1 Hz induced changes in nuclear membrane irregularities, hyperchromasia, and abnormal chromatin distribution.

The expression of early osteogenic differentiation marker genes (i.e., *ALPL*, *RUNX*-*2*, and *COL1A1*) and the ALP enzymatic activity, whose functionally mirrors the differentiation grades of osteoblastic cells, have been reported to be involved in metastasis in osteosarcoma patients [36]. Our data demonstrated that both genes and the enzyme were not observed to be sensitive to the 24 h 1 Hz stretching stimulation (Figure 4C,D), indicating that the mechanically induced morphology changes we observed do not seem to be linked to a modulation of the differentiative osteoblast phenotype. In any case, we cannot exclude that there could be an effect on the late osteogenic differentiation process after longer treatments.

Since cell volume regulation (a fundamental aspect of the homeostasis of the cell) and nuclear size dysregulation have been reported to affect cell migration by altering gene regulation, cell signaling, and function [27,42], we looked at the induced mechanical change in SAOS-2 cell motility capacity. Interestingly, we observed that the 24 h duration of the stimulus was long enough to profoundly change two relevant functions that influence the metastatic potential of osteosarcoma cells in vitro, namely, (i) cell motility and (ii) cell adhesion. On the one hand, the adhesion capability of mechanically pre-treated cells decreased by around 30% compared with untreated control cells (Figure 5C,D). On the other hand, mechanically pre-treated cells could migrate and transmigrate faster than their control counterparts (Figure 5A,B).

Despite the fact that the mechanisms and types of cell migration and invasion have been described and studied relatively well, there are currently no highly efficient and validated molecular markers for the identification of migrating/invading tumor cells in tumors and, therefore, for the assessment of their invasive potential [29,43]. In this regard, the differences in the complexity of boundary protrusions and cell peripheral roughness (Appendix A and Figure 3) between 24 h 1 Hz treated versus untreated cells suggest it would be useful to focus future research on these regions in order to identify possible new markers for highly migratory cells.

Besides the specific mechanical signals that have been reported to regulate cancer invasion and migration, the soluble extracellular context is also crucial for osteosarcoma cell biology in influencing metastatic cell journeys [42]. Mechanical stresses from the microenvironment have been reported to regulate the PI3K/Akt signaling pathway (the most activated downstream effectors in the oncogenic landscape), promoting the expression of MMP-2 and MMP-9, thus degrading ECM and enabling osteosarcoma cell invasion and metastasis [29,30,31]. Several MMPs have been associated with mature invadopodia [38], including MMP-2, which was found to have an upregulated expression following mechanical stimulation [44].

Although we did not observe any significant change in the total levels of Akt protein following mechanical stimulation, when the secretion of MMP-2 was examined using gelatin zymography of the conditioned medium, we found that 1 Hz mechanical stimulation was able to upregulate its enzymatic activity (Figure 6). Despite the fact that mechanical stresses have been reported to affect MMP levels [30] (Figure 6), which, in turn, contribute to cell migration, invasion, and metastasis [45], the 24 h 1 Hz induced upregulation of the motile capacity of the SAOS-2 cells was found to be independent of the activity of any type of MMP, given that both migration and transmigration assays were unaffected by the presence of ilomastat (a broad-spectrum inhibitor of MMPs) (Figure 7). Itho et al. report that MMP-mediated cell migration could alternatively occur either through proteolytic or non-proteolytic mechanisms [46]. However, our results indicate that the stretch-induced upregulation of migration does not happen through a proteolytic mechanism.

On the contrary, GsMTx4, a specific mechanosensitive stretch-activated ion-channel inhibitor, effectively counteracts the cell migrative upregulation induced by 24 h 1 Hz stimulation (Figure 7E). Recently, the GsMTx4 peptide has also been reported to inhibit mechanically induced epithelial–mesenchymal transitions in keratinocytes [47]. Indeed, the control gate function of ion stretch channels represents the best-known mechanosensory regulation that is induced by the tension forces of cell membranes [40].

Considering that (i) the increase in membrane tension can activate specific mechanosensitive ion channels, including Piezo and transient receptor potential (TRP) channels [48,49,50,51]; (ii) mechanical compression has been reported to enhance the invasive phenotype of breast cancer cells [52]; and (iii) TRP and/or Piezo channels have been shown to play an essential role in the regulation of cell migration [24], it is reasonable to hypothesize that the 24 h 1 Hz stimulation stretches the cell membrane, possibly activating TRP and/or Piezo channels.

Understanding the underlying mechanisms of mechano-sensation and developing therapeutic applications for oncology treatments are current and future research goals. This study proves (though a single cell line) that some of the metastatic potential properties of osteosarcoma cells (such as the nuclear-to-cytoplasmic (N/C) ratio, cell shape, peripheral cell rugosity, cell adhesion, and cell migration) are mechanoresponsive.

From an oncological perspective, we believe that 24 h 1 Hz exogenous stimulation could be exploited to identify the cellular pathways underlying the biological abnormalities in malignant cells. These mechanobiology studies may represent an alternative approach to identifying cell signaling and new markers for the highly migratory cells that drive osteosarcoma progression and metastasis.

## 4. Materials and Methods

### 4.1. Cell Culture

Human SAOS-2 (HTB-85) cells were purchased from the American Type Culture Collection (Rockville, MD, USA). The cells were cultured in Dulbecco’s Modified Eagle’s Medium (4.5 g/L glucose)/Ham F12 (1:1) (Invitrogen, Carlsbad, CA, USA) supplemented with 10% fetal bovine serum (FBS) (Euroclone s.p.a.,MI Milano, Italy), Penicillin–Streptomycin Solution 100X (Gibco, Life Technologies, Carlsbad, CA, USA), and Amphotericin B 100X (Biowest, Riverside, MO, USA) at 37 °C in an atmosphere of 5% CO_2_. The culture medium was changed twice a week, during which nonadherent cells were discarded.

### 4.2. Mechanical Stretch Application

A silicone well culturing plate (CellScale Biomaterials Testing, Waterloo, ON, Canada) is made of a deformable silicone rubber that displays a 10A Shore (on the Shore D hardness scale), which allows the culturing system to follow deformation applied (along the *x*-axis) by the mechanical stretch MechanoCulture FX device (CellScale Biomaterials Testing, Waterloo, ON, Canada [53]). The silicone wells were coated with rat type I collagen (Enzo Life Sciences, Farmingdale, NY, USA) at a 50 µg/mL concentration in PBS solution. SAOS-2 cells were trypsinized and seeded at a density of 2 × 10^5^ cells per well (culture area: 16 wells, each well 8 mm × 8 mm). For each dataset, two silicon plates were seeded, and both were exposed to the same experimental conditions, but just one of the two was subjected to the mechanical stretch, while the spare one was used as a static unstimulated control. At 24 h after cell seeding, the mechanical stretch was applied to cells using the MechanoCulture FX mechanical stimulation system (CellScale Biomaterials Testing, Waterloo, ON, Canada) for 1 or 3 days. The MCFX software (https://www.cellscale.com/products/mcfx/, accessed on 1 February 2023) was used to program the device with the phases, cycles, and sets of choices (Appendix A). The strain regimen was performed under the selected stretch, i.e., 0.5% elongation applied at 1 Hz cyclical frequency for 24 h, alternating 1 h of stretching and 3 h of rest (for further information, see Appendix A). To measure the cell response of the adherent live cells immediately after the mechanical stretch, a cell-based experimental setup on a silicone deformable plate was developed to detect fluorescence and spectrometry on a multi-plate reader (see Appendix A).

### 4.3. Cell Count and Protein Content

Immediately after the mechanical treatment, the stretched samples and control counterparts were processed as follows. The cell density was determined by counting the cells after trypsinization with an automated cell-counting chip^TM^ TECAN spark^®^ multimode reader (Tecan Group, Männedorf, Switzerland). A CellTiter 96^®^ AQ_ueous_One Solution Cell Proliferation Assay (MTS) (Promega, Milano, Italy), the 3-(4,5-dimethylthiazol-2-yl)-2,5-diphenyl-tetrazolium bromide (MTT) colorimetric method (Merk Life Science s.r.l., Milano, Italia), and a CyQUANT™ NF Cell Proliferation Assay (Thermofisher Scientific Inc. Waltham, MA, USA) were used to count cells in a population and to measure proliferation in a microplate format (Appendix A). Protein content was measured in extracts of confluent cells grown on silicone well plates. The SAOS-2 cells were then detached from the plates and centrifuged. The supernatant was discarded, rinsed with PBS, and then lysed with RIPA buffer containing 150 nM NaCl, 50 mM Tris-HCl, 0.1% SDS, 1% Triton X-100, 1% Na cholate, 1% NaOV, 1 mM NaF, 1 mM EDTA, PMSF 1X, and Protease Inhibitor Cocktail (Cell Signalling Technology, Danvers, MA, USA). Protein quantifications were performed on the protein extracts using a Bradford assay (BioRad, Hercules, CA, USA). The protein content of cell lysates was determined with the Biorad protein assay and with bovine serum albumin (BSA) as a standard. Optical density was measured at 595 nm with a TECAN spark^R^ multimode reader (*Tecan* Group *Ltd.*, Männedorf, Switzerland).

### 4.4. Cell Viability and Proliferation Assay

Rigid supports, which we customized on the flexible silicone well plate, were built to allow for fluorescence and spectrophotometric reading via the multi-plate TECAN spark^R^ reader (*Tecan* Group *Ltd.*, Männedorf, Switzerland). The 3D model was designed with the e-Drawing program (version 30.1.0.0032; 1999–2022 Dassault Systèmes Corporation). As a 3D-printing filler, polylactic acid was used (see Appendix A).

#### 4.4.1. MTS Assay on Flexible Silicone Plates

Cell viability was determined through a CellTiter 96^®^ AQ_ueous_One Solution Cell Proliferation Assay (MTS) (Promega, Milano, Italy) in the silicone well plate. Cells were seeded on a silicone plate at a density of 400 cells/mm^2^ and cultured in the growth medium as previously indicated. After mechanical treatment, 20 μL of MTS-PMS solution was added per well followed by 2 h of incubation at 37 °C with 5% CO_2_. Then, the absorbance of the formazan products of each well was recorded at 490 nm using an Infinite^®^200 PRO multi-well plate reader (Tecan Group Ltd., Männedorf, Switzerland).

#### 4.4.2. Cell Viability Assay on 96-Well Plates

To examine cell proliferation ability, cells were seeded on silicone plates at a density of 110 cells/mm^2^ and cultured in the growth medium as previously indicated. The cell number at each time-point was counted with a 3-(4,5-dimethylthiazol-2-yl)-2,5-diphenyl-tetrazolium bromide (MTT) colorimetric assay (Merk Life Science s.r.l., Milano, Italy). Briefly, 20 μL of MTT solution (5 mg/mL in PBS *w*/*w* Ca^2+^ and Mg^2+^) was added to each well followed by 2 h of incubation at 37 °C with 5% CO_2_. Then, 100 μL of extraction buffer (5% SDS in N,N-Dimethylformamide) was added to each well followed by 2 h of incubation at 37 °C with 5% CO_2_. In total, 150 μL per silicone well was transferred to a 96-well plate to measure the optical density of the formazan products. The absorbance of the formazan products was recorded at 570 nm using an Infinite^®^ 200 PRO multi-well plate reader (Tecan Group Ltd., Männedorf, Switzerland) (Appendix A).

### 4.5. Quantitative RT-PCR Analysis

The impact of the mechanical treatment on gene expression was evaluated as follows: following the 24-h mechanical stretch, treated SAOS-2 cells and the static control counterpart were trypsin-detached, pelleted, processed, and -analyzed for the following target genes: *COL1A1*, *RUNX-2*, *BGLAP*/*OCC*, and *ALPL*. RNA extraction from cellular pellets was prepared using the TRIZOL Reagent (Roche Diagnostics GmbH, Mannheim, Germany), according to the manufacturer’s protocol. RNA quality was examined by measuring the absorbance ratio at 260 nm and 280 nm through the NanoQuant Plate of an Infinite^®^200 PRO multi-well plate reader (Tecan Group Ltd., Männedorf, Switzerland). RNA was reverse-transcribed with SensiFAST™ cDNA Synthesis Kit (Bioline, Meridian Bioscience, London, UK) following the manufacturer’s specifications. Gene expression was measured using iTaq Universal SYBR Green Supermix (Biorad Laboratories, Hercules, CA, USA). qRT-PCR was performed using a LightCycler 96 Real-Time PCR System (Roche Diagnostics GmbH). For data analysis, the expressions of all genes were normalized using the ΔΔ cycle threshold method against human glyceraldehyde 3-phosphate dehydrogenase *GAPDH* gene expression. The sequences of primers used are as follows.

GeneSequences (5′-3′)
*h ALPL F*
ACCTCGTTGACACCTGGAAG
*h ALPL R*
CCACCATCTCGGAGAGTGAC
*h RUNX-2 F*
GAGTGGACGAGGCAAGAGTT
*h RUNX-2 R*
AGCTTCTGTCTGTGCCTTCTG
*h COL1A1 F*
GTGCGATGACGTGATCTGTGA
*h COL1A1 R*
CGGTGGTTTCTTGGTCGGT
*h BGLAP/OCC F*
CACTCCTCGCCCTATTGGC
*h BGLAP/OCC R*
CCCTCCTGCTTGGACACAAAG
*h GAPDH F*
AGAAGGCTGGGGCTCATTT
*h GAPDH R*
AGGGGCCATCCACAGTCTT

### 4.6. Alkaline Phosphatase Activity Assay on Silicone Plate

The ALP activity was performed within the silicone plate using an alkaline phosphatase blue microwell substrate containing Bromo-Chloro-Indolylphosphate (BCIP^®^) (Sigma-Aldrich, Chemical Co., St. Louis, MO, USA) (Appendix A). Prior to any reaction with alkaline phosphatase, the BCIP^®^ reagent is a colorless to faint blue solution, and the Nitro Blue Tetrazolium (NBT/Thiazolyl Blue/Nitro BT) reagent is a yellow solution. The two components, when mixed, develop a bluish-purple product when reacting with alkaline phosphatase in microwell-type assays. The colorimetric reaction was assessed by using the TECAN spark^R^ multimode reader to measure the absorbance of the product at 520 nm within the first 10 min of incubation at 37 °C. The absorbances at t 10 min were normalized for the absorbances recorded at t 0.

### 4.7. Adhesion Assay

Right after the mechanical treatment, the stretched samples and control counterparts were trypsin-detached, pelleted, and processed (Appendix A). A cell adhesion assay after mechanical stimulus was performed on a 96-well plate that had been precoated with rat tail collagen I at a 50 µg/mL concentration and maintained at 37 °C for 2 h. SAOS-2 cells were seeded in DMEM high glucose at a density of 400 cells per well and maintained at 37 °C for 24 h under conventional tissue-growing conditions. The following day, the cells were washed and cell live confluence was recorded with the live imaging program of a TECAN spark^R^ multimode reader (*Tecan* Group *Ltd.,* Männedorf, Switzerland). Cell monolayers were then crystal-violet-stained through 10 min of incubation in 0.5% crystal violet dissolved in 20% *v*/*v* methanol solution at 37 °C (Sigma Chemical Co., St. Louis, MO, USA). After several washes to remove the excess crystal violet, pictures were taken by an Olympus CKX53 inverted microscope equipped with an EP50 Microscope Digital Camera (Olympus Life Science, Waltham, MA, USA). Cell counts were based on counts of ten fields from digital images taken randomly at ×4 magnification by the ImageJ software ImageJ bundled with 64-bit Java 8 (ImageJ bundled with 64-bit Java 8, http://rsb.info.nih.gov/ij/, accessed on 1 February 2023).

### 4.8. Morphological Microscopy

The measurement of the areas, perimeters, lengths, roughness, and heights was performed on subconfluent cells. Ab initio, to validate the SAOS-2 cell morphology, AFM and confocal microscopy were performed on cells seeded at a 110 cells/mm^2^ density on conventional glass coverslips precoated with collagen I cultured at 37 °C for 24 h and then morphologically inspected.

Furthermore, immediately after the mechanical treatment, the stretched cells and their control counterparts were trypsin-detached, pelleted, counted, and seeded at 110 cells/mm^2^ density on conventional glass coverslips precoated with collagen I cultured at 37 °C for 24 h and then morphologically inspected (Appendix A).

Similarly, on the silicone plate, SAOS-2 cells were seeded at low density (110 cells/mm^2^), cultured at 37 °C for 24 h, and treated or not with a mechanical uniaxial stretch. Adherent cells were fixed with 4% paraformaldehyde in phosphate-buffered saline (PBS) for 20 min at room temperature and then washed twice with bi-distilled water. The silicone wells were cut out, rinsed, and glued to a tailored glass coverslip for the AFM or confocal microscope inspections as described in the following paragraphs (Appendix A).

#### 4.8.1. Nuclear Morphometric Analysis via Atomic Force Microscope (AFM)

AFM measurements were performed with the microscope working in the repulsive regime of contact mode in air at room temperature, as previously described [54]. Bruker silicon nitride MSNL-10 cantilevers were employed. Constant force images with a force of 1 nN were acquired with a typical scan rate of 2–4 s/row (800–1600 points/row). Data were background-subtracted and then analyzed using the Gwyddion software. Nuclear shape was quantified by the eccentricity, E (according to Equation (1)). A high value for the E index indicates a round shape, while a low value refers to an elongated nuclear form. The nuclear surface was calculated from measurements of the major and minor axes according to Equation (2). Differentiation between areas corresponding to the nucleus and off-nucleus was based solely on topographical features. While it is assumed that areas marked as the “nucleus” indeed have a predominant height contribution from the nucleus of the cell, we cannot discard possible contributions to the measured height from other parts of the cell. However, for simplicity, we simply refer to these areas as the nucleus and off-nucleus. Cell height corresponds to the average value of the nucleus and off-nucleus areas of each measured cell. The roughness of SAOS-2 cells was calculated using AFM images, restricting the analysis to the surface areas near the cell edge (within 2 μm of the cell boundaries). The cell periphery was defined by those areas within approximately 2 μm of the cell perimeter, as shown in Figure 4A). An average of the topographical roughness was then calculated for this area. Similar definitions of cell periphery have been used in past studies of cell morphology with AFM techniques [55].

#### 4.8.2. Confocal Microscopy

An Olympus LEXT OLS 4000 confocal microscope (Olympus Corporation, Tokyo, Japan) was used in confocal acquiring mode; ×20 or ×50 objective lenses were used (NA = 0.60, WD = 1 mm and NA = 0.95, WD = 0.35 mm, respectively) with a 405-nm laser. The images were acquired using the microscopy laser–confocal method without any DIC filter, with a size of 1024 × 1024 pixels. Using the ×20 lens, the xy area imaged was 648 × 648 μm, and using the ×50 lens, the xy area imaged was 258 × 258 μm, thus giving a lateral spatial resolution of 0.025 μm per pixel. The LEXT-specific files were exported in the jpg image format. The quantitative geometrical shape analysis for the identification of the stretch-induced morphological differences was performed as previously described [22]. Areas, distances, and perimeters measured on cells and nucleus sizes were normalized with a reference scale using ImageJ software ImageJ bundled with 64-bit Java 8 (http://rsb.info.nih.gov/ij/, accessed on 1February 2023). The number of pseudopodia per cell was counted, considering the cell processes with a measurement of more than 15 μm. The cell elongation was quantified not only by eccentricity, E, but also through roundness, R (Equation (1) and Equation (3), respectively). In both cases, a high value for these indexes indicates a round shape, while a low value reflects an elongated shape. Confocal microscopy pictures were inspected and analyzed to determine the N/C ratio: the whole cell surface and nuclear area were measured for each cell individually. The measurements of the cell area and the nuclear area were performed employing freehand and elliptical selections, respectively (ImageJ software). The circularity, C, shape descriptor was derived according to Equation (4). High values for C correlate with the absence of a star-like shape.

### 4.9. Western Blot

Immediately after the mechanical treatment, the stretched samples and control counterparts were trypsin-detached, pelleted, and processed as follows. To assess the protein expression level, SAOS-2 samples were lysed with RIPA buffer containing 150 nM NaCl, 50 mM Tris-HCl, 0.1% SDS, 1% Triton X-100, 1% Na cholate, 1% NaOV, 1 mM NaF, 1 mM EDTA, PMSF 1X, and a Protease Inhibitor Cocktail (Cell Signaling Technology, Danvers, MA, USA). From 30 to 60 µg of proteome was loaded per lane into precast gels (Mini-PROTEAN^®^ TGX™ Precast Gels 4–20%, BioRad, Hercules, CA, USA) and then transferred to a PVDF membrane (Amersham, Buckinghamshire, UK). The membrane was blocked with 5% milk (Sigma Chemical Co., St. Louis, MO, USA) and probed with specific primary and secondary antibodies. The blots were developed by using Enhanced Chemiluminescence (ECL) detection systems (Amersham, UK). The following primary antibodies were used: *GAPDH* (dilution 1:10,000) (GeneText Irvine, Irvine, CA, USA, GTX100118) as a control, Phospo-Akt (Ser473) (D9E) (dilution 1:500) (Cell Signaling Technology, Danvers, MA, USA, #4060), and Akt (pan) (11E7) (dilution 1:1000) (Cell Signaling Technology, Danvers, MA, USA, #4685). The intensities of bands were measured through an image analysis program (ImageJ, public domain software; NIH, Bethesda, MD, USA) and quantified using a scale of arbitrary units (AU). The results obtained were analyzed using statistical software (Prism 9 ver. 9.0.0 (121), GraphPad Software).

### 4.10. Wound Healing Migration Assay

Following the 24 h mechanical treatment, treated SAOS-2 cells and the static control counterpart were seeded (at a density of 1 × 10^4^ cells per well) on conventional culturing 96-well plates and incubated at 37 °C with 5% CO_2_ for 24 h to allow a cell-confluent monolayer to firmly adhere to the rat type I collagen precoated bottom. Then, the 100% cell confluence monolayer was scratched to create a gap with a 10 µL pipette tip and rinsed with PBS to remove the suspended cells. Following the introduction of a scratch, cells were then incubated with serum-free DMEM supplemented or not with the indicated 1 μM GsMTx-4 (ab141871, Abcam, Cambridge, UK) or 1,6 μM ilomastat (CC10, Sigma-Aldrich Chemical Co., St. Louis, MO, USA) and were maintained at 37 °C with 5% CO_2._ The wound closure was monitored for up to 48 h, and the gaps were photographed at 0 h and 20 h after scratching. The area of scratches was calculated using microscope measuring instruments (Olympus CKX53 equipped with an EP50 Microscope Digital Camera; Olympus Life Science, Waltham, MA, USA) and cell confluence (Tecan Group Ltd., Männedorf, Switzerland). The wound migration was expressed as wound closure (WC)% and as relative wound density (RWD) calculated according to Equations (5) and (6), respectively, as previously reported [56]
(5)$$Wound\;Closure\;\%=\frac{A\left(t=0\right)-A(t\Delta h)}{A(t=0)}$$
where

*A*(*t* = 0) is the area of the wound measured immediately after scratching at *t* = 0.

And

*A*(*t*∆*h*) is the area of the wound measured h hours after the scratch is performed.
(6)$$\%RWD\left(t\right)=\frac{w\left(t\right)-w(0)}{c\left(t\right)-w(0)}$$
where

*w*(0) is the cell density of the wound area measured at time 0 immediately after the scratch is created.

*w*(*t*) is the cell density of the wound area measured at time *t*.

*c*(*t*) is the cell density of the cell area measured at time *t*.

Areas were calculated using the ImageJ public domain software (NIH, Bethesda, MD), whereas cell density was measured using the spark multi-mode plate reader (Tecan Group Ltd., Männedorf, Switzerland, Life Sciences).

### 4.11. Cell Transmigration Assay

The cell migration assay was assessed using a trans-well Boyden Chamber. After mechanical treatment, in serum-free DMEM, supplemented or not with the indicated compound, 1 μM GsMTx-4 (ab141871, Abcam, Cambridge, UK) or 1.6 μM ilomastat (CC10, Sigma-Aldrich Chemical Co., St. Louis, MO, USA), SAOS-2 cells were trypsin-detached and 4 × 10^4^ cells in 100 μL of serum-free DMEM (supplemented or not with the compounds mentioned above) were seeded in the upper chamber of a trans-well 24-well plate (8.0 μm pore size, Corning-Costar Corporation, Cambridge, MA, USA). After 10 min of incubation at 37 °C, 600 μL of culture medium containing 10% FBS was added to the lower chamber as a chemoattractant. After 24 h at 37 °C with 5% CO_2_ for incubation, the trans-well chambers were removed. The cells that had migrated to the bottom of the wells were photographed with an optical microscope digital camera (Olympus CKX53 equipped with an EP50, Olympus Life Science, Waltham, MA, USA). Migrated cells were 0.1% crystal-violet-stained and subjected to a microscopic inspection. Cell counts were based on counts of ten fields from digital images taken randomly at ×4 magnification by the ImageJ software. The percentage of migrated cells was calculated using the average migrated cells in stretched group/average migrated cells in the control group, ×100%.

### 4.12. Zymography

Gelatin substrate zymography was used for the evaluation and characterization of the SAOS-2-cell-conditioned media, as previously described [57]. Right after the mechanical treatment, the conditioned medium from the stretched samples and control counterparts were harvested and processed as follows: the confluent cell layers on the silicone plates were washed three times with a room-temperature PBS buffer. Media were replaced by serum-free DMEM, and cells were incubated at 37 °C overnight (mechanically treated or not). The cell-conditioned media were concentrated 10-fold via speed-vac centrivapor. Protein concentrations were determined using a Bradford assay (BioRad, Hercules, CA, USA) and confirmed by running SDS-PAGE gel electrophoresis in 10% polyacrylamide gels prior to zymography. Concentrated medium samples were diluted in an SDS-polyacrylamide gel electrophoresis sample buffer under nonreducing conditions without heating. Samples were separated by 10% SDS-polyacrylamide gels, which were co-polymerized with 1 mg/mL gelatin type B (Sigma Chemical Co., St. Louis, MO, USA). After electrophoresis runs, gels were washed twice for 30 min in 2.5% Triton X-100 and incubated overnight in an “activity buffer” (50 mmol/L Tris-HCl, pH 7.5, 10 mmol/L CaCl_2_, 150 mmol/L NaCl) at 37 °C. As a control, an additional gel was incubated with an activity buffer that was supplemented with 1 mM of metalloproteinase inhibitor (GM6001), which completely abolishes the appearance of the clear transparent band. Gels were stained in Coomassie Blue R 250 (Bio-Rad, Milano, Italy) in a mixture of methanol: acetic acid: water (4:1:5) for 1 h and destained in the same solution without dye. Gelatinase activities were visualized as distinct bands, indicating the proteolysis of the substrate. The intensities of the gelatinolytic activity areas were measured through an image analysis program (ImageJ public domain software, NIH, Bethesda, MD, USA) and quantified using a scale of arbitrary units (AU). The results obtained were analyzed using statistical software (Prism 9 ver. 9.0.0 (121), GraphPad Software). Recombinant human MMP-2 (R&D Systems, Boston, MA, USA) was used as a reference standard for proMMP-2.

### 4.13. Data Analysis

The results are the means ± SEM, and differences between means were determined with the parametric *t*-test using the GraphPad Prism 9.01 software (San Diego, CA, USA).

## Figures and Tables

**Figure 1 ijms-24-07686-f001:**
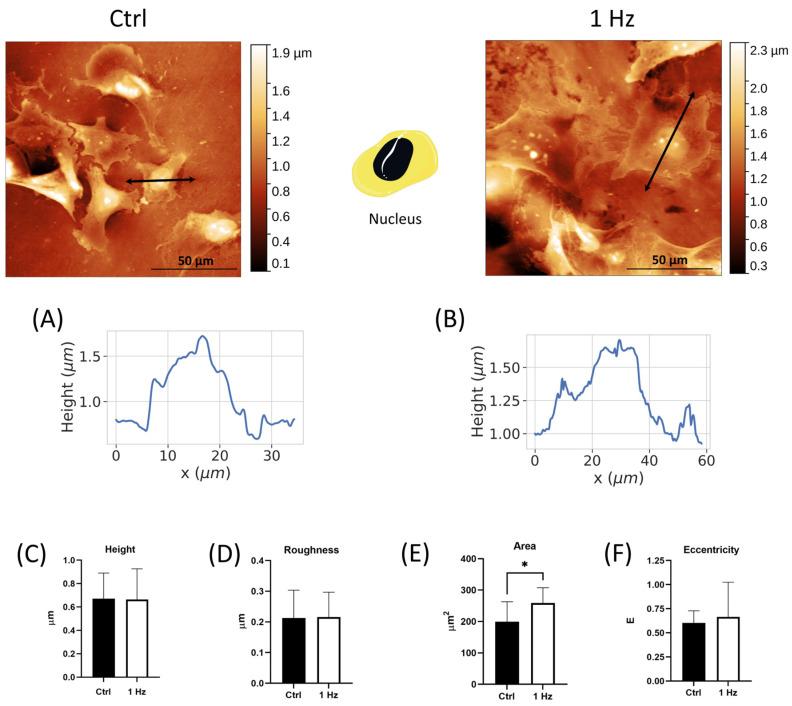
AFM analysis of the nucleus of SAOS-2 cells on a silicone plate treated or not with a 24 h 1 Hz uniaxial cyclic stretch (along the *x*-axis). Upper panel: cell drawing—the cell region where the measurements were taken is in black (i.e., the nucleus) (https://smart.servier.com, accessed on 1 February 2023), and there are two representative AFM pictures of untreated and stretched cells. The double-pointed arrows indicate the positions where the cell profiles were measured. Panels (**A**,**B**) report the cell profiles for untreated and stretched cells, respectively. From Panel (**C**) to Panel (**F**), the comparison of four parameters between untreated cells (black) and 24 h 1 Hz treated cells (white) (error bars represent standard error of the mean). Panel (**C**) displays the nuclear heights. Panel (**D**) shows the nuclear surface roughness. Panel (**E**) displays nuclear areas, and Panel (**F**) reports the nuclear eccentricity. AFM analyses were performed on two stretched and two unstretched samples with at least 16 cells per sample preparation. Student’s *t*-test was used for statistical analysis, and the results are shown as the mean ± SD. * *p* < 0.05; treated cells compared with control cells.

**Figure 2 ijms-24-07686-f002:**
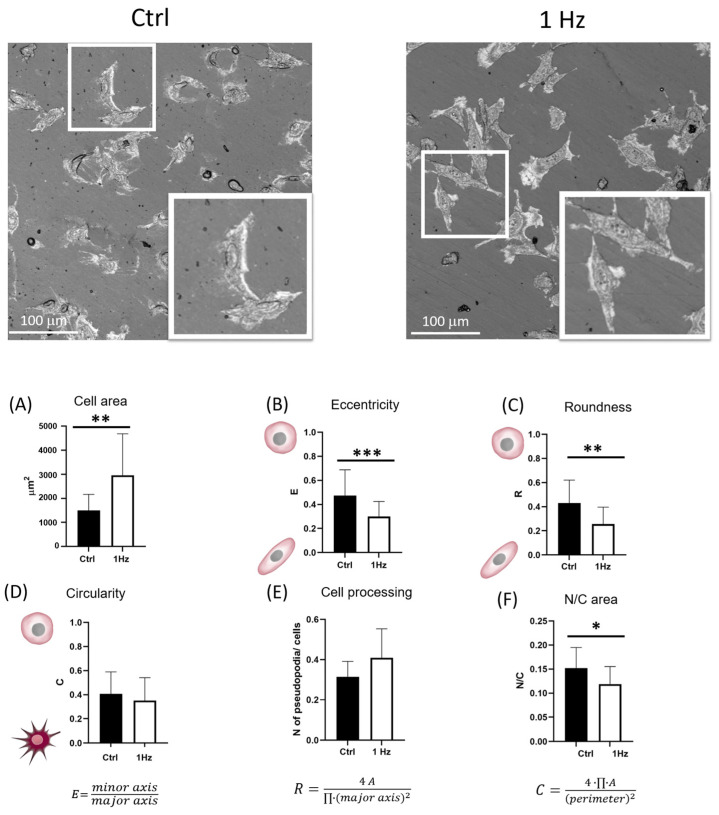
Morphological changes were induced in the whole cells by the 24 h 1 Hz cyclic stretch (along the *x*-axis). The upper panel shows two representative confocal microscopy pictures of treated and control SAOS-2 cells on a silicone plate. From Panels (**A**–**F**), the column bar graphs report the plot means and the SD and display the difference in the mean of the cyclically stretched and unstimulated SAOS-2 cells. Panel (**A**) plots the cell area. Panel (**B**) refers to eccentricity E. Panel (**C**) displays the roundness index R. Panel (**D**) shows the circularity index C. Panel (**E**) displays the amount of cell processing counted per cell. Panel (**F**) reports the ratio between the area of the nucleus and its cytoplasm in the same cell (N/C). ImageJ 1.52 was employed for image analysis, and a paired Student’s *t*-test was used for statistical analysis. The analysis was performed on 3 biological replicates with at least 18 cells per condition. The results are shown as the mean ± SD. * *p* < 0.05, ** *p* < 0.01, and *** *p* < 0.001; treated specimens compared with control static cells.

**Figure 3 ijms-24-07686-f003:**
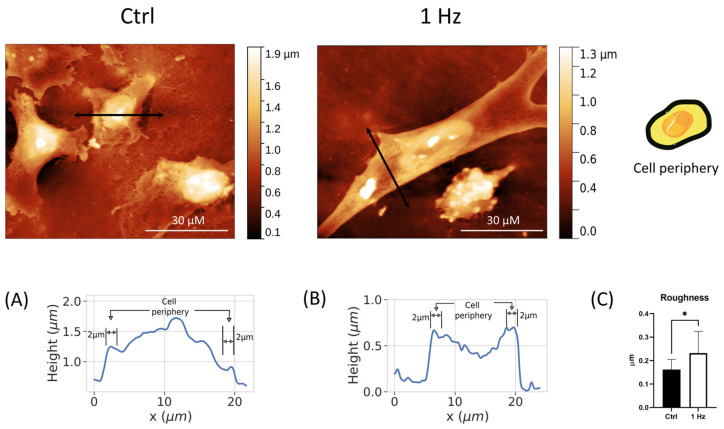
AFM roughness changes in the peripheral cell induced by the 24 h 1 Hz cyclic stretch (along the *x*-axis) measured on silicone plates. Upper panel: cell drawing—the cell periphery of cells is represented in black (https://smart.servier.com, accessed on 1 February 2023), and there are two representative AFM pictures for untreated and stretched cells. The double-pointed arrows indicate the positions where the cell profiles were measured. Panels (**A**,**B**) report the cell height profiles for untreated and stretched cells, respectively, to better show the two-micron peripheral area (for further details, see Appendix A). Roughness was measured with respect to the top surface of the cells. Panel (**C**): Comparison between roughness averages of the peripheral cell surfaces. Black histogram represents untreated cells and white histograms represent 1 Hz 24 h treated cells (error bars represent standard error of the mean). AFM analyses were performed on two stretched and two unstretched samples with at least 16 cells per sample. Student’s *t*-test was used for statistical analysis, and the results are shown as the mean ± SD. * *p* < 0.05; treated cells compared with the control cells.

**Figure 4 ijms-24-07686-f004:**
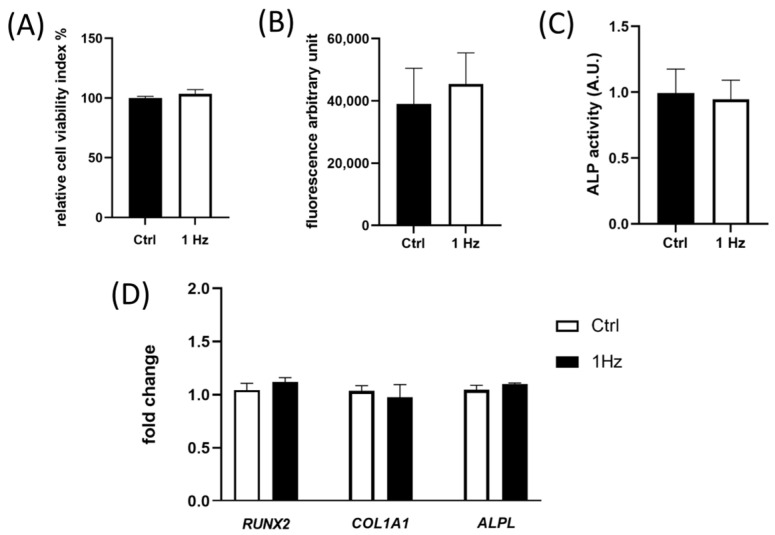
Cell-based assays on a silicone plate of adherent SAOS-2 cells. The 1 Hz 24 h stretch stimulation impacted neither cell viability nor the osteoblastic properties of SAOS-2 cells. The comparison between measurements displays black histograms for untreated cells and white histograms for 1 Hz 24 h treated cells (error bars represent the standard error of the mean). (**A**) Indirect cell viability quantitation via soluble spectrometric MTS probe; (**B**) cell viability derived via DNA quantification based on Cy Quant fluorogenic probe; (**C**) on-plate detection of ALP activity using a Blue ALP kit (blue microwell substrate containing BCIP^®^); (**D**) the impact of a 24 h 1 Hz uniaxial stimulation on the gene expression of three pro-osteogenic differentiative markers (i.e., *RUNX-2*, *COL1A1*, and *ALPL*). Student’s *t*-test of the histograms of Panels (**A**–**D**) showed no significant differences between the treated cells compared with the control static cells. Statistical analyses were performed on three biological replicates with at least three technical replicates per condition.

**Figure 5 ijms-24-07686-f005:**
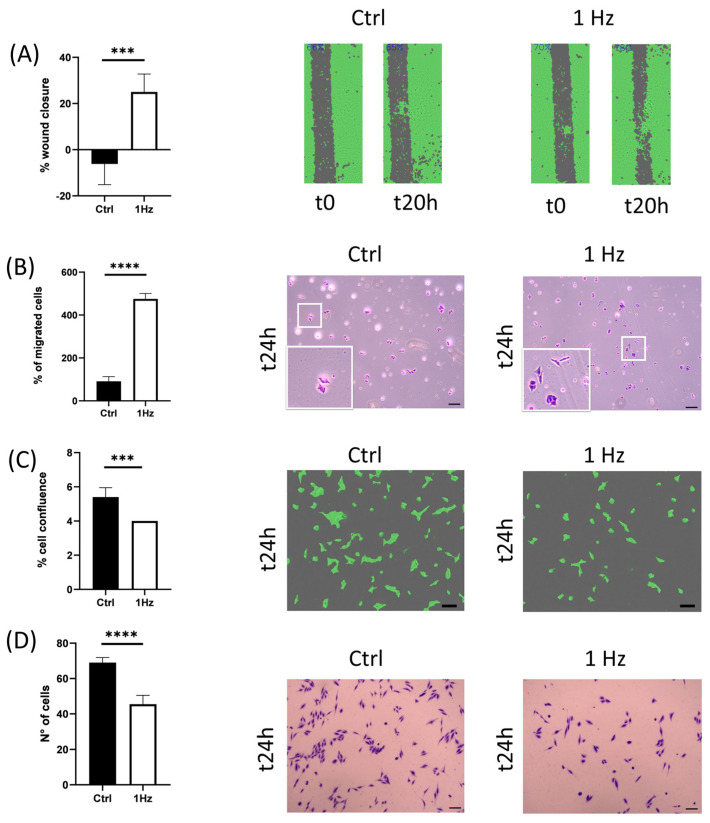
Cell motility capacity was found to be upregulated by the 24 h 1 Hz stretch stimulation (on a plastic support) (Panels (**A**,**B**)). Panel (**A**) shows a comparison of the 20 h cell migration between treated and untreated SAOS-2 cells using a scratch test. Panel (**B**) evaluated the cell transmigration comparison between the control and stretched cells using Boyden chambers. The 1 Hz cyclical stretch reduced the cell-adhesion capacity of SAOS-2 cells (Panels (**C**,**D**)). Panel (**C**) reports cell confluence measured for both conditions after 24 h of attachment. Panel (**D**) reports the cell counts measured after 24 h of attachment to precoated wells (see Materials and Methods for further details). The statistical analyses were performed on three biological replicates with six technical replicates for Panel (**A**,**C**), four technical replicates for Panel (**B**), and five technical replicates for Panel (**D**). Scale bar: 100 µm. An unpaired *t*-test analysis was adopted to calculate the significant difference. The results are shown as the mean ± SD. *** *p* < 0.001, and **** *p* < 0.0001; treated cells compared with control static cells.

**Figure 6 ijms-24-07686-f006:**
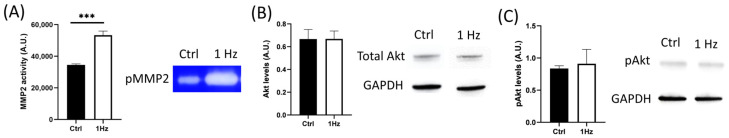
Panel (**A**): The zymography displays the effect of cyclical stimulation on the gelatinolytic activity of the MMP-2 pro-enzyme (pMMP-2) found in the 24 h conditioned media of treated or untreated SAOS-2 cells. A representative image of gel zymography is reported. Panel (**B,C**): Western blot analysis of cell extracts treated or untreated with 24 h 1 Hz stimulation. A representative image of a Western blot for the Akt protein is reported. Filters were probed with specific antibodies for the total Akt (60 kDa), Phospo-Akt(Ser473) (60 kDa),and *GAPDH* (37 kDa). A densitometric analysis of the gel band areas was performed using the ImageJ free processing software and quantified using a scale of arbitrary units (error bars represent the standard error of the mean). An unpaired *t*-test analysis was used to calculate the significant differences. The results are shown as the mean ± SD of treated cells compared with control static cells (*** *p* < 0.001). The statistical analyses were performed on three biological replicates with six technical replicates: three technical replicates for Panel (**A**) and three technical replicates for Panel (**B,C**).

**Figure 7 ijms-24-07686-f007:**
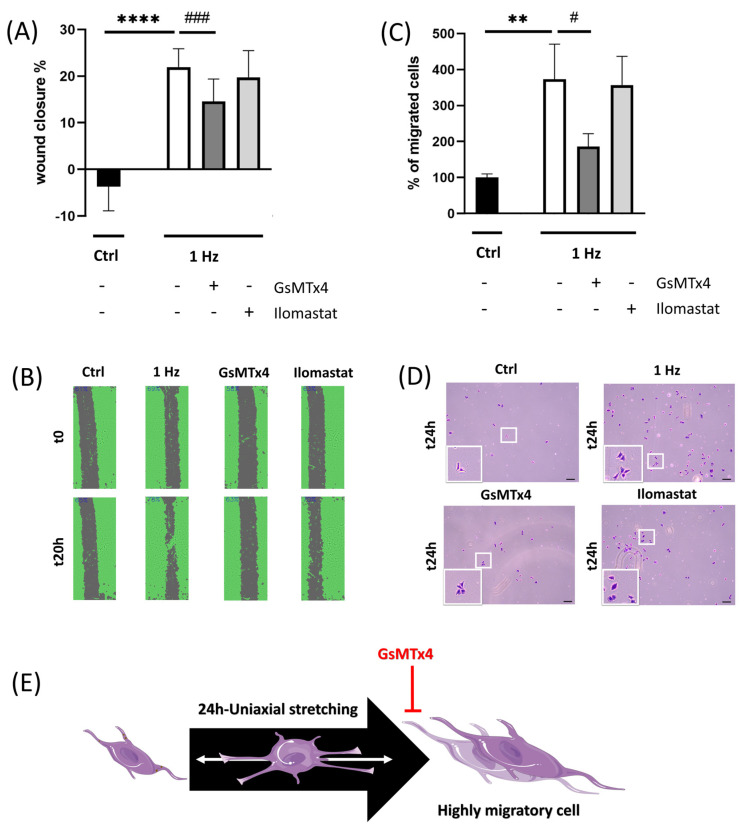
The 24 h 1 Hz stretch-induced upregulation of motile cell capacity was inhibited by stretch-activated ion-channel inhibitors (on plastic supports) (Panels (**A**–**D**)). Panel (**A**): A comparison of cell migration between stimulated and unstimulated SAOS-2 cells, treated with GsMTx4 or ilomastat, was assessed using a scratch assay. Panel (**B**): Representative images of migrated cells after the 24 h 1 Hz mechanical stretch and treated with GsMTx4 or ilomastat. The area of scratches was calculated as a percentage using Tecan spark instruments (Tecan Group Ltd., Männedorf, Switzerland). Panel (**C**): A comparison of cell transmigration between control and stretched cells, subsequently treated with GsMTx4 or ilomastat, was made using a Boyden chamber. Panel (**D**): Representative images of transmigrated cells stained with crystal violet after the 24 h 1 Hz mechanical stretch and treated with GsMTx4 or ilomastat (scale bar: 100 µm). Panel (**E**): Sketch of the GsMTx4 inhibition of the mechanically induced, highly migratory transition (https://smart.servier.com, accessed on 1 February 2023). An unpaired *t*-test analysis was adopted to calculate the significant difference. The statistical analyses were performed on three biological replicates with twelve technical replicates for Panel (**A**) and four technical replicates for Panel (**C**). The results are shown as the mean ± SD. ^#^
*p* < 0.05 ** *p* < 0.01, ^###^
*p* < 0.001, and **** *p* < 0.0001; treated cells compared with control static cells (see Materials and Methods for further details).

## Data Availability

All data generated or analyzed during this study are included in this published article and its Appendix A.

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
