# Peer review of "Cyclic Stretch-Induced Mechanical Stress Applied at 1 Hz Frequency Can Alter the Metastatic Potential Properties of SAOS-2 Osteosarcoma Cells"

_ijms, 2023, doi:10.3390/ijms24097686_

Round 1

Reviewer 1 Report

Dear Authors

Thank you for your valuable paper, it is suitable for publication in its current form.

Author Response

We thank Referee 1 for considering our paper to be valuable and suitable for publication in its original form.

Reviewer 2 Report

1.The quality of the pictures needs to be improved.

2. Why choose 20h for migration and transmigration test after 24h-1Hz treatment?

3. pAKT should be detected as well.

Reviewer 3 Report

The study is investigating mechanobiological response of osteosarcoma SAOS-2 cells. The idea is good but the realization is not the best. The data interpretation should be checked and corrected according to the presented micrographs and/or other results. Additional silicon surface controls are needed in order to answer to many pending questions.  

Remarks:

1.       The introduction part is missing the explanations, why MMPs were chosen to measure. Actually, everywhere the authors write MMPs, while the only MMP2 was measured. Similar explanation about the ion channels and AKT is also missing.

2.       Figure 1. Somehow seems that magnification of two micrographs is different. The authors should check it.  

However, the interpretation of comparison of stretch and not stretch cells saying that stretch cells became bigger is not correct. It is obvious that stretched cells adhere better than the not stretch cells. Seems that the stretched silicon surface become more cell-attachment friendly than not stretched. The authors should do additional surface controls.

3.       The same problem is with the Figure 2. The control cells do not feel good on the not stretched silicon surface – many of them are dead and without the nucleus! The cells on the stretched surface feel much better. If the cell viability measurement on both surfaces (stretched and not stretched is similar) that means that something is very wrong with the AFM sample preparation.  However, the cell viability measurements tendency confirms that cells on not stretched surface feel worse than on stretched.

             The authors measured many cell morphological parameters but everything is not the cell morphology but cell attachment dependent, i.e. the cell elongation and other “stretched” cell changes are due to their better attachment to the tensile softened surface compare to the regular one.

4.        Fig. 3. Sorry, but the magnification of those two micrographs again is different (even 30um is of different magnification), again the cell attachment is different and all other changes also are better cell attachment on the tensile softened surface related process.  

             Probably the authors could rather talk about the roughness of attachment surface than cells. Moreover, the interpretation that better attached cells have higher roughness is more than strange.

             The picture of “cut avocado” is also not clear.

5.       The same problem might be with the migration assay – if the cells better attached to the stretch softened surface, they also migrate much better. The quality of Fig. 5 B, C, D is very low and should be improved.

The Fig. 5 B micrographs does not show that stretched cell confluence is higher than not stretched. It is strange that the transmigration of stretched cells is lower than not stretched if their migration was better?

6.         Fig. 6. Which forms of MMP2and AKT were identified? Phosphorylated or not? The total level of protein for MMP2 is not shown.

7.       Fig. 7. The time points of A, C, D are not clear. The quality of Fig. 7 micrographs should be improved.

8.       Conclusions are superficial and not concrete.

Minor remarks.

1.       The title has a word “Title”, which does not belong to it.

Reviewer 4 Report

In this study, Alloisio et al. investigated the effect of cyclic stretch-induced mechanical stress on an osteosarcoma cell line SAOS-2 in terms of metastatic potential. The authors found that 24 hr cyclic stretch-induced mechanical stress induces changes in nuclear size, and several properties of cell shapes including cell size, and roundness. The authors then examined phenotypical changes by the mechanical stress. They found no impact on cell viability and expression levels of pro-osteogenic marker genes. On the other hand, they found upregulation of cell motility along with downregulation of cell adhesion. The authors examined MMP-2 activity by the mechanical stress and found that MMP2 activity was enhanced, along with no changes in Akt expression levels. The authors examined the impact of MMP inhibitors, and found no effects on the mechanical stress-induced enhancement of cell motility. On the other hand, the authors found that inhibition of ion channel inhibited the stress-induced enhanced cell motility.

I have several questions and comments to this study.

1. Rationale for some experiments was quite unclear.

The authors showed enhanced MMP2 activity by the mechanical stress, while treatment with MMP inhibitors showed no effect on its effect on cell motility. On the other hand, the authors sudden picked up ion-channel inhibitors and found its significant effect on cell motility. Why the authors chose MMP2 among a number of extracellular proteinases? Why the authors examined ion channel inhibitor? The authors have to show the experimental evidence and clarify rational of these experiments.

The authors also examined Akt signaling pathway by western blotting. The rationale for focusing on Akt is not quite clear, and the authors need to explain rationale. Additionally, total protein levels of Akt cannot tell whether it is activated or not. The authors have to show phosphorylation levels of it if they wanted to understand activation of Akt signaling pathway.  

2. Related to question #1, in the last experiment the authors showed potential effect of mechanical stress on ion channel-mediated cell motility. If so, how did the mechanical stress affect ion channel (what type of ion-channel was involved in? was the expression levels upregulated?)?

3. The authors examined the effect of the mechanical stress on cell motility. Given that the assay time takes more than24 hr (figure 5 and 7), it cannot exclude the effect on cell proliferation. However, in the main text, the authors mentioned the effect on cell viability only. The authors should differentiate the effect on cell proliferation and cell motility.

4. The authors examined nuclear shape by AFM. How can it detect nuclei specifically without using specific nuclei markers?

5. Is the tile correct?

Round 2

Reviewer 3 Report

The authors did some improvements. However, many inaccuracies still left, some authors responses could be more constructive and better scientifically based. The interpretations about the better stretched cells attachment compared to the control cells was not taken into consideration. The methodological as well as result parts still need strong correcting.

1.      The authors response concerning the cell attachment: “Answer 2c) The referee’s speculations might be right. However, our data show that 24h-1Hz treatment decreases SaoS-2 cell ability to adhere to conventional culturing conditions in vitro.“ contradicts to data in Figure 2A showing the “cell area”, which is the same as “cell attachment area” since it is adherent culture. It also means that the attachment of stretched cells is better compared to the not stretched cells on silicon surface. The AFM micrographs also confirms better attachment of stretched cells than control ones, i.e. the control cells worse survived sample preparation.  Otherwise, the different cell attachment to silicon surface should be confirmed.

2.      The data of Fig. 3. There is no explanation what the different cell periphery “roughness” means. Actually the height of the cell periphery has different meaning than roughness.  

3.      The AFM nuclear and cell periphery analyzing Figures can be combined in to one, clearly stating where are nucleus and where are the cell periphery measurements, particularly when AFM micrograph of control cells is the same, just cropped in Fig. 3 showing higher magnification.

4.      Fig.1, 2, 3 legends do not have statistic of how many cells were measured. It is not enough to measure just one cell many times.

5.      The AFM micrographs and calculations rise doubts since the cell attachment is different, which affects the cell nucleus and whole cell size measurements.

Moreover, the cells for the AFM were fixed on glass (Figure S2). It is interesting, how the cells were stretch on the glass?The legends of AFM measurements should clearly state the surface type.

6.      Fig. 5. The micrograph (B and C) quality is still very low and not acceptable. The legend should clearly explain panels C and D, i.e. are those panels of cell transmigration data? If yes, they contradict with the data in panel B. The surface type is not identified.

7.      Fig. 7 panel D. Quality is slightly better than in previous Fig.5. but still not the best. Legend needs explanation what migration is shown in panel D. The migration and transmigration as well as cell attachment surfaces should be clearly stated.  

8.      The Method part. Why the method “5.4.2 Cell viability assay on 96-well plates” was used if the authors state that everything was done on silicon plates with or without stretch? Unless the cell viability was done on 96 well plate and compared with the silicon plate, which is wrong. In all legends the control should be clearly explained.

9.      The same question is about the section ” 5.6.1 On a 96-wells- plate ALP assay“.

10.  The method description says that AFM as well as confocal measurements were done both on plastic and silicon surfaces. So, on which surface the cells are in presented AFM micrographs? The legends do not have an explanation on which surface the experiments were done.

The unused methods descriptions should be eliminated.

11.  The descriptions of „Wound healing migration assay“, „Cell transmigration assay“ and „zymography“  methods need corrections, since it is not clear how the stretch of the cells was done if the cells were seeded on plastic surface.

Reviewer 4 Report

The authors answered all questions. I'm satisfied.

Author Response

We are glad that Referee 4 is satisfied with the round1 version of the manuscript, and we all thank Referee4 for her/his valuable suggestions.

Round 3

Reviewer 3 Report

The unclear methodological and result parts of the manuscript have been improved and more clearly explained. The Fig. 5C micrographs still need bars.

Author Response

We would like to thank all Referees (particularly Referee 3) for their fruitful suggestions.